# Use of Laryngeal Mask and Anesthetic Management in Hamadryas Baboons (*Papio hamadryas*) Undergoing Laparoscopic Salpingectomy—A Case Series

**DOI:** 10.3390/vetsci10020158

**Published:** 2023-02-15

**Authors:** Annalaura Scardia, Pietro Laricchiuta, Marzia Stabile, Claudia Acquafredda, Luca Lacitignola, Annamaria Uva, Antonio Crovace, Francesco Staffieri

**Affiliations:** 1Ph.D. Course in “Tissues and Organs Transplantations and Cellular Therapies”, D.E.O.T., University of Bari, 70121 Bari, Italy; 2Section of Veterinary Clinics and Animal Production, D.E.O.T., University of Bari, 70123 Bari, Italy; 3Zoosafari, 72015 Fasano, Bari, Italy; 4Section of Veterinary Internal Medicine, DiMeV, University of Bari, 70010 Bari, Italy

**Keywords:** baboon, laryngeal mask, anesthesia, laparoscopy

## Abstract

**Simple Summary:**

In this observational study, we aimed to describe the cardiorespiratory variations during laparoscopic abdominal surgery in baboons (*Papio hamadryas*) anesthetized via a laryngeal mask. Animals were immobilized with a combination of tiletamine/zolazepam and medetomidine. A laryngeal mask was inserted to allow for the delivery of oxygen and isoflurane. Cardiovascular variables showed no alterations, while the respiratory parameters indicated a rise in respiratory rate and end-tidal carbon dioxide concentration during the pneumoperitoneum. Hypercapnia was mild, even with the animal in spontaneous breathing, and it was solved rapidly after the discontinuation of the pneumoperitoneum. The results of this study suggest that laparoscopy can be performed safely in spontaneously breathing baboons. Furthermore, adequate ventilation and gas exchange can be achieved using a laryngeal mask without the need for endotracheal intubation.

**Abstract:**

The study aims to describe the anesthetic and airway management of baboons *(Papio hamadryas)* undergoing laparoscopic salpingectomy with a laryngeal mask airway (LMA) device. Eleven baboons received tiletamine-zolazepam and medetomidine; anesthesia was induced with propofol. An LMA was positioned for oxygen and isoflurane administration in spontaneous respiration. Heart rate (HR), mean arterial pressure (MAP), respiratory rate (RR), end tidal carbon dioxide (EtCO_2_), minute volume (MV), and peripheral hemoglobin oxygen saturation (SpO_2_) were recorded before (PREPP) and immediately after abdomen insufflation (PP1), at 10 (PP2), 20 (PP3), and 30 (PP4) minutes during pneumoperitoneum, and after (POSTPP) pneumoperitoneum. The respiratory rate was significantly higher at all times compared to PREPP. The end tidal carbon dioxide concentration was significantly higher at PP2, PP3, PP4, and POSTPP, compared to the previous times. The higher values for RR and EtCO_2_ were registered at PP4: 22.7 (95% CI 17.6–27.8) breaths/min and 57.9 (95% CI 51.9–63.8) mmHg, respectively. The minute volume was significantly higher at PP4 and POSTPP compared to the other times. The higher value for MV was registered at POSTPP (269.1 (95% CI 206.1–331.8) mL/kg/min). This protocol is suitable for baboons undergoing laparoscopic salpingectomy. The LMA was easy to insert and allowed for good ventilation, gas exchange, and delivery of the anesthetic in spontaneous breathing baboons.

## 1. Introduction

The Hamadryas baboon (*Papio hamadryas*) is a terrestrial, medium sized monkey from the Old World monkey family. The males are silver colored with a pronounced mantle; they can weigh approximately 18–20 kg, and the females weigh 9–11 kg. Females are capeless and brown, and weigh 9–11 kg, and in the wild, they reach sexual maturity at four years of age; estrous cycles last approximately 30 days, and during ovulations, they have a pronounced genital swelling [1]. Laparoscopic surgery is very common in veterinary practice. Salpingectomy is reported as a sterilization method in non-human primates (NHPs) [2], and recently, a salpingectomy laparoscopic technique in baboons has been described by our group [3].

The perioperative systemic stress response, as assessed via the determination of metabolic and endocrine indices, is significantly reduced in the laparoscopic surgery, compared to open surgery [4,5,6].

Laparoscopy requires adequate anesthetic management, which must prevent the major side effects of pneumoperitoneum. The infusion of carbon dioxide to distend the abdomen promotes the cranial displacement of the diaphragm and causes an impairment of the pulmonary functional residual capacity [7]. In addition, the use of CO_2_ to induce the pneumoperitoneum causes the absorption of the gas from the peritoneal cavity, with a consequent increase in the partial pressure of arterial carbon dioxide (PaCO_2_) [8]. These are the main reasons for why adequate airway management and a close monitoring are of great importance during laparoscopy. Cardiovascular impairment is another side effect of pneumoperitoneum, due to the rise of intrabdominal pressure (IAP) and consequently, the reduction in venous return.

Moreover, the anesthetic management of NHPs has several critical and specific aspects, including adequate intramuscular immobilization, airway management, and smooth and rapid recovery [9]. Most of the studies available in the literature regarding the anesthesia of baboons refer to experimental procedures [10,11,12]. Sometimes, for their complexity and invasiveness, they have required the use of neuromuscular blocking agents such as pancuronium and succinylcholine [13,14,15]. None of these were performed for clinical purposes, and this is a limitation because few data regarding the anesthesiologic management of baboons are available. Laparoscopy is reported in baboons in the literature for experimental purposes [16,17] but its cardiovascular and respiratory effects in a clinical scenario are missing.

Endotracheal intubation is the principal technique for airways management and the prevention of aspiration in NHPs anesthesia. The anatomy of the high airways of primates is different compared to other species. A large part of the tongue lies in the hypopharynx because of the descent of the larynx, and this can restrict the visualization of the entrance and the diameter of the larynx. Furthermore, the trachea is smaller and shorter than in other species of equivalent weight, increasing the risk for mono-bronchial intubation [9,18]. Supraglottic airway devices (SADs), such as the laryngeal mask airway (LMA), form a seal around the larynx, usually with an inflatable cuff, providing the management of the airways intermediate between the facemask and endotracheal intubation, in terms of anatomic position, invasiveness, and safety [19]. The use of LMA is reported in NHPs such as chimpanzees, gorillas, and a gibbon [20,21], but no mention exists for baboons.

This case series is aimed at describing the anesthesiologic management of 11 baboons undergoing laparoscopy, in which an LMA has been used for airway management. We describe the changes of the physiological parameters induced using pneumoperitoneum, and our experience related to the use of an LMA for airways control.

## 2. Materials and Methods

The study was approved by the Ethical Committee for Clinical studies of the Dipartimento dell’Emergenza e dei Trapianti di Organi (D.E.O.T.) of University of Bari “Aldo Moro” (Approval num: 05/2020). Surgical procedures were performed by an experienced veterinary surgeon (>100 laparoscopies per year) and the zoo veterinary team. The study was performed from November 2020 to December 2020.

Twelve adult and sub-adult female baboons weighting 4–14.4 kg and undergoing elective laparoscopic salpingectomy were enrolled in the study. Before surgery, abdominal ultrasonography was performed to evaluate the reproductive tract and the presence of pregnancy. Animals in pregnancy or endometrial disorder (hyperplasia, endometritis, and neoplasia) were excluded from the sterilization program.

With the help of the zoo’s keepers, a harem was isolated from the whole group in a separated environment three days before the planned surgery. Each animal was individually recognized via microchip number. Animals were kept fasting for 15 h, and water was withheld 8 h before surgery.

The animals were restrained in a squeeze cage and hand injected with tiletamine-zolazepam (Zoletil 100 mg/mL, Virbac S.r.l., Milan, Italy) and medetomidine (Domitor 1 mg/mL, Zoetis s.r.l., Rome, Italy) IM combined in the same syringe. The individual dose of the drugs for immobilization was calculated based on the body weight approximation and the experience of the veterinarian of the zoo, at an estimated dose of 2 and 4 mg/kg tiletamine-zolazepam and 20–60 μg/kg medetomidine. Accurate weights of animals were obtained once the animal was deeply sedated, and the accurate dose received was then calculated. A 22G intravenous cannula was placed in the cephalic vein and used for the administration of drugs and Lactate Ringer solution. Venous (cephalic vein) blood samples were collected, stored in ice, and analyzed within 12 h for a complete blood count (CBC) and biochemical profile. Just before induction, peripheral hemoglobin oxygen saturation (SpO_2_, %) was measured with the animal breathing room air (0.21 inspired Oxygen Fraction, FiO_2_), using a transmission pulse oximeter placed on the tongue. The SpO_2_ value was recorded after at least 1 min of monitoring with an adequate plettismographic signal, as assessed based on the observation of the trace (SpO_2_ Air Test, SpAT) [22].

Propofol (PropoVet, Zoetis Italia S.r.l., Rome, Italy) IV was administered to effect until optimal jaw relaxation was achieved to allow for the insertion of a laryngeal mask (LMA Supreme ™, The Laryngeal Mask Company Limited, Le Rocher, Victoria, Mahe, Seychelles) of an adequate size. With the animal in dorsal recumbency and the tongue pulled rostrally, gentle pressure was applied to the palate with the LMA, while moving the device caudally until resistance was felt. The size of LMA was chosen based on the manufacturer’s recommendations for human weights (size 1: <5 kg, size 2: 10–20 kg), and adjusted if necessary. After insertion, the cuff was inflated according to the manufacturer’s recommendations, and the LMA was secured. The criteria for the evaluation of the adequacy of the size and positioning of the LMA were: the absence of leaks at 20 cm H_2_O of airway pressure, and the adequacy of the capnographic curve. The leak test was performed, closing the adjustable pressure-limiting (APL) valve of the breathing circuit, turning off the oxygen flow, and squeezing the reservoir bag until 20 cm H_2_O of airway pressure was reached, and checking for the absence of a gross air leak (hearing gas flow and/or smelling isoflurane). If the criteria of positioning were not achieved after the first LMA placement, the cuff was deflated and a second attempt was made. If the second attempt was not satisfactory, endotracheal intubation was performed.

Animals were connected to a rebreathing circuit in spontaneous ventilation, and isoflurane (IsoFlo^®^, Zoetis Italia S.r.l., Rome, Italy) in oxygen was administered for maintaining anesthesia. The vaporizer was set to 1.0–1.5%. A mainstream capnograph (EMMA^®^ capnograph, Masimo Corporation 52 Discovery, Irvine, CA, USA) was positioned between the LMA connector and the anesthesia circuit to obtain the respiratory rate (RR, breaths per minute) and the end tidal carbon dioxide concentration (EtCO_2_, mm Hg). A multiparametric monitor (Compact 5, medical ECONET, Im Erlengrund 20, Oberhausen, Germany) was used to monitor the heart rate (HR, beats per minute) and noninvasive blood pressure (NIBP, mmHg). A spirometer was connected to the expiratory limb of the anesthetic system to measure the tidal volume (Vt, mL) and the minute volume (MV, L/min) (RM121, Medishield B.V., Molengraaffsingel 12–14, 2629 JD Delft, The Netherlands). A pulse oximeter positioned on the tongue was used to assess SpO_2_. The adequacy of the depth of the anesthesia was based on the assessment of the palpebral reflex, jaw tone, and the variation of hemodynamic parameters. All animals received as a preoperative standard treatment meloxicam (Metacam, Boehringer Ingelheim, Milan, Italy) 0.2 mg/kg IV and cefazoline (Cefazolina Teva, Teva Italia S.r.l., Milan, Italy) 20 mg/kg IV.

When the animal reached a stable plane of anesthesia, surgery was initiated. Before the insufflation of the abdomen with CO_2_, the following parameters were recorded twice within a 5 min interval: HR, Systolic, Mean, and Diastolic Blood Pressure (SAP, MAP, and DAP, mm Hg), EtCO_2_, RR, Vt, MV, and SpO_2_. A mean value for the two measurements was calculated and reported as PREPP. The same recordings were made immediately after abdomen insufflation (PP1), at 10 (PP2), 20 (PP3), and 30 (PP4) minutes during pneumoperitoneum, and five minutes after the abdomen was deflated (POSTPP). The maximum IAP reached was 8 mm Hg.

In the case of a sudden increase in HR, RR, and blood pressure, a bolus of fentanyl 2 µg/kg (Fentadon, Dechra Veterinary Products, Turin, Italy) IV was administered as a rescue analgesia.

Hypotension was defined as a MAP that was lower than 60 mmHg, and it was treated by reducing (if possible) the dose of the inhalant anesthetic, and administering fluid boluses as needed. In non-responsive cases, dopamine (5–9 mcg/kg/min) was infused.

Hypercapnia was defined as an EtCO_2_ value of higher than 55 mmHg, and manually assisted positive pressure ventilation was delivered in the case where one or more of the following conditions were observed: EtCO_2_ > 65 mmHg, RR < 5 bpm, and SpO_2_ < 95%.

Ivermectin (Ivomec, Boehringer Ingelheim Animal Health Italia S.p.a., Milan, Italy), 0.2 mg/kg SC, and amoxicillin (Betamox LA, Vétoquinol Italia S.r.l., Bertinoro, Italy) 25 mg/kg IM were administered at the end of the surgery.

At the end of the procedure, isoflurane was discontinued, and the animals were positioned in lateral recumbency. When the palpebral reflex returned, the cuff was deflated and the LMA was removed. A new SpAT was performed as previously described, then the animals were moved into a warmed recovery area. They were monitored until consciousness returned and they were able to stand. The time from the discontinuation of isoflurane to standing was recorded.

Normal distribution was assessed using the Shapiro–Wilk test. Mean value and 95% confidence interval (95% CI) were calculated for all data. Physiologic parameters were compared among study times using the one-way ANOVA test, in order to evaluate any significant modification related to the pneumoperitoneum. *p* < 0.05 was considered as statistically significant.

## 3. Results

Of the 12 animals included, one was found to be pregnant after sedation, and thus, the procedure was not performed. The 11 remaining baboons completed the study without complications. The mean body weight of the animals was 9.6 kg (CI 95% 7.3–11.9).

The dose of tiletamine-zolazepam used ranged between 2.3 and 10.4 mg/kg (mean 4.4 mg/kg, CI 95% 2.8–6.1), and that of medetomidine was between 0.01 and 0.04 mg/kg (mean 0.02 mg/kg, CI 95% 0.017–0.03). The mean values of the hematologic parameters, CBC, and biochemical profile, are reported in Table 1 and Table 2, respectively.

The LMA insertion was successful in all cases, and no animals required intubation.

The mean anesthesia time was 36.5 min (CI 95% 30.5–42.5). The mean value and 95% CI of HR, MAP, RR, EtCO_2_, TV, MV, and SpO_2_ of the baboons included in the study are reported in Table 3. None of the animals required the rescue dose of fentanyl.

There was a statistical difference of some parameters between the study times. The respiratory rate was significantly higher at all times compared to PREPP, and at PP4, compared to all of the previous times. The higher value of RR was registered at time PP4 (Figure 1c). EtCO_2_ was significantly higher at PP2, PP3, PP4, and POSTPP, compared to times PREPP and PP1 (Figure 1e). The minute volume was significantly higher at PP4 and POSTPP compared to the other times. The higher value for MV was registered at POSTPP time (Figure 1d). The heart rate was significantly lower at PP1 compared to PREPP (Figure 1a). The mean arterial pressure was significantly higher at PP2 compared to PP1 (Figure 1b). TV and SpO_2_ did not show any significant variations during the times of the study.

Five of eleven baboons had episodes of transitory hypotension during pneumoperitoneum that were responsive to the reduction in isoflurane delivery and fluids.

Hypercapnia occurred from PP2 to POSTPP, and the higher value was registered at PP4 58 (52–64). However, none of the cases required assisted manual ventilation.

The SpAT measured prior to induction ranged between 86% and 99% (mean of 93%, 95% CI 88–98), and that measured post-LMA removal was between 93% and 99% (mean of 97%, 95% CI 95–100). The data referring to SpAT are not available for all of animals included in the study; subjects number 4 and 9 missed pre-anesthesia SpAT, number 1 and 10 missed post-anesthesia SpAT, and numbers 6, 7, and 11 missed both pre- and post-anesthesia SpAT.

At 52 min (95% CI 20.1–83.9) after isoflurane was discontinued, the animals were able to stand. The animals experienced mild ataxia and incoordination during recovery.

## 4. Discussion

General anesthesia for laparoscopic salpingectomy in hamadryas baboons, managed with a laryngeal mask in spontaneous ventilation, was feasible in all cases of the study and did not show any major complication. The LMA proved to be safe and adequate in ensuring oxygenation, CO_2_ elimination, and the delivery of anesthetic gas (Figure 2). The adequacy of the size of the LMA to the laryngeal opening of the baboons suggests that the manufacturer’s recommendations for human patients are also valid for this species.

The hematological values are within the reference range reported for the species, unless the BUN, which is slightly higher. Although the data show the situation at a single time point, they are useful for verifying the health status of the animals, as well as the hepatic and renal functions, and the CBC.

Although laparoscopy is reported in different species of NHPs [2,24], few data exist regarding the effect that pneumoperitoneum has on the physiological variables of the anesthetized animals. In baboons, the effect of laparoscopy is reported in experimental studies [16], showing a rise in RR and a decrease in HR, with an IAP of between 7 and 26 mmHg, while respiratory distress occurred after 30 min of IAP of 22 mm Hg. An intrabdominal pressure of up to 77 mm Hg was not lethal in most of the subjects [17], while in pregnant baboons, maternal and fetal physiologic alterations were seen at 20 mm Hg IAP [16].

The major concerns about the anesthetic event during laparoscopy are related to the insufflation of the abdomen. The pneumoperitoneum is associated with the reduction of incentral venous return, and impaired ventilation. Hypotension is a clinical alteration related to the reduction in venous return. In this study, MAP decreased at PP1 (73 ± 22 mmHg) compared to PREPP (82 ± 15) after the induction of pneumoperitoneum, but the difference was not statistically significant. The significant increase in MAP at time PP2 (85 ± 24 mmHg) was probably attributable to the basic sympathetic activation related to the surgical stimulus. In none of the cases was the level of anesthesia superficial or was rescue analgesia required. Although the mean value of MAP was above 60 mmHg, 5 of 11 baboons experienced temporary hypotension (range of MAP 45–58 mmHg) during pneumoperitoneum, which was treated by adjusting the anesthetic plan and/or via the administration of a fluid bolus.

Hypercapnia occurred during penumoperitoneum in this study. It is considered as a normal occurrence during laparoscopy for two reasons: the absorption of CO_2_ from the abdominal wall, and the impairment of ventilation caused by the reduction in lung functional residual capacity and compliance. For this reason, mechanical ventilation is usually recommended [8]. In this study, all animals were in spontaneous breathing and there was no need for assisted ventilation during pneumoperitoneum, based on the predetermined cut-off conditions. We observed a progressive increase in MV and RR that was concomitant with the rise of EtCO_2_ (MV reached its higher value at POSTPP, 269.1 ± 93.5 mL/kg/min, and RR at PP4, 23 ± 8 breaths/min). This compensatory mechanism successfully contributed the avoidance of an excessive rise of EtCO_2_. Performing blood gas analysis would be ideal to evaluate ventilation and to exclude the presence of a large P(a-Et)CO_2_, indicating a mismatch between ventilation and perfusion. TV and SpO_2_ were stable during the study.

In the literature, all protocols reported for the anesthetic management of baboons refer to experimental studies [10,11,12]. There are no data about the anesthesiologic management of baboons in clinical practice. The combination of TZ and M used for sedation was already reported by Fahlman to achieve immobilization in other species of primates for routine clinical procedures, although different doses were used, especially for TZ (0.9–2.3 mg/kg) [25]. The combination of MTZ induced a rapid and smooth induction, good muscle relaxation, and stable RR and HR, but decreases in SAP and temperature over time and low SpO_2_ values were observed [25]. In our study, we are missing data about body temperature, and regarding the arterial blood pressure, the administration of propofol, isoflurane, and IV fluid therapy makes a comparison between the two studies impossible. However, a low SpO_2_ at room air was a common finding (88–92%), which in our study was solved by O_2_ supplementation (FiO_2_ > 0.8) that started after the LMA was correctly inserted. The administration of oxygen during the anesthetic procedure improved SpO_2_ intraoperatively (97–98%), as well as in the immediate postoperative period when the animals were breathing room air (SpAT 97 ± 2%).

The recovery was smooth, with light signs of ataxia and incoordination. The variability of the times of standing may be due to a different dosage of the drugs received for sedation, since a wide range in doses of TZM and weights of the animals are observed.

A laryngeal mask was introduced in human medicine in the 1980s from an idea by Dr. Brain [26]. In veterinary medicine, its use has been reported in dogs [27], cats [28], rabbits [29], pigs [30], capibaras [31], and in NHPs (chimpanzees, gibbons, and gorillas) [20,21]. The results obtained from NHPs have been encouraging, suggesting LMA as a suitable tool for airway control and for administering volatile anesthetics. Cerveny compared an endotracheal tube (ETT) with LMA in gorillas, and found a higher SpO_2_ and partial pressure of oxygen (PaO_2_), and a lower RR in the LMA group. However, the treatment group was not randomized, and animals with a greater risk of respiratory complications were included in the group ETT, and so this may have accounted for the observed differences. In addition, manual ventilation, when necessary, was performed successfully in animals ventilated with LMA [21]. In this study, administering oxygen through an LMA allowed for adequate ventilation and oxygenation, as defined by the SpO_2_ and EtCO_2_ values. SpO_2_ increased from pre-anesthetic SpAT (93%, 95% CI 88–98) when oxygen started to be administered through LMA (97–98% during the times of the study). Even the EtCO_2_ increased significantly during pneumoperitoneum, and it remained high even after. Anyway, the rise of EtCO_2_ was not clinically severe, and it lay below the threshold value for starting manually assisted ventilation.

In humans, LMA apported some advantages over the endotracheal tube in elective laparoscopic procedures, and in patients without specific requirements or diseases: the maintenance of adequate oxygenation and ventilation perioperatively, and the minor incidence of postoperative pharyngolaryngeal morbidity, from a minimum to zero incidence of hemodynamic stress response, and minor rates of postoperative nausea and vomiting [32,33]. A study comparing LMA (LMA ProSeal) and ETT in infants refers to easier insertion and airway management in groups receiving LMA [34], and these results were confirmed by Asida and Ahmed [35], with a high percentage of success of insertion (94.4%) between the first and second attempts. In the study from Cerveny et al., the authors mentioned that the device was of easy and quick insertion compared to ETT in gorillas, although statistically significant data are missing [21]. The ease of insertion of LMA could play an important role in the spreading of this kind of device in emergency settings, and among less experienced professionals. Moreover, in the anesthesia of NHPs, the spread of the use of the supraglottic devices can decrease the risk for monobronchial intubation due to the shorter trachea compared to humans.

One of the major disadvantages of the use of LMA is the lack of airway protection in case of regurgitation. Although an appropriate fasting time before anesthesia was mandatory for all baboons, the induction of pneumoperitoneum could have been an event promoting regurgitation. In order to limit the risk of aspiration, a second-generation LMA has been used (LMA Supreme ™, Figure 3).

This type of LMA has a drainage tube for the gastric tube passage, to prevent the distention of the stomach from liquids and gases, and to protect against regurgitation and aspiration. In addition, a higher sealing pressure is reported with LMA Supreme ™ compared to LMA classic (34.6 (3.1) cm H_2_O vs. 26.1 (2.1) cm H_2_O) [36]. Thanks to these enhanced safety features over the first-generation devices, second-generation SADs are more adequate for use in obese patients, obstetrics patients, and for laparoscopic surgeries in humans [37]. Even if in this study, the gastric tube was not inserted preventively, it can be considered in the case of the observation of gastric reflux.

Among the limitations of the study, there are the lack of comparison with endotracheal intubation, and the wide range of doses of the sedative drugs, that could have influenced the physiological parameters throughout the procedure.

## 5. Conclusions

The cases described show that the anesthetic management used in this observational study is suitable for baboons undergoing laparoscopic salpingectomy. Respiratory rate, EtCO_2_, and MV increased during pneumoperitoneum, although the level of hypercapnia remained quite mild. The LMA was easy to insert and allowed for adequate ventilation, even during pneumoperitoneum, and the delivery of anesthetic in spontaneous breathing baboons. 

## Figures and Tables

**Figure 1 vetsci-10-00158-f001:**
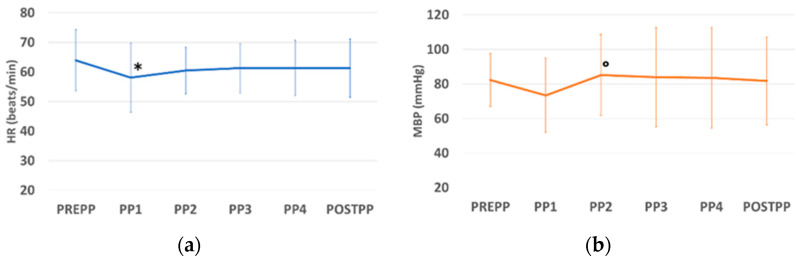
The figures show the mean ± standard deviation (SD) of HR (**a**), MAP (**b**), RR (**c**), MV (**d**), and EtCO_2_, (**e**) during the times of the study. * = *p* < 0.05 compared to PREPP. # = *p* < 0.05 compared to PP4. ° = *p* < 0.05 compared to PP1. § = *p* < 0.05 compared to PREPP, PP1, PP2, and PP3.

**Figure 2 vetsci-10-00158-f002:**
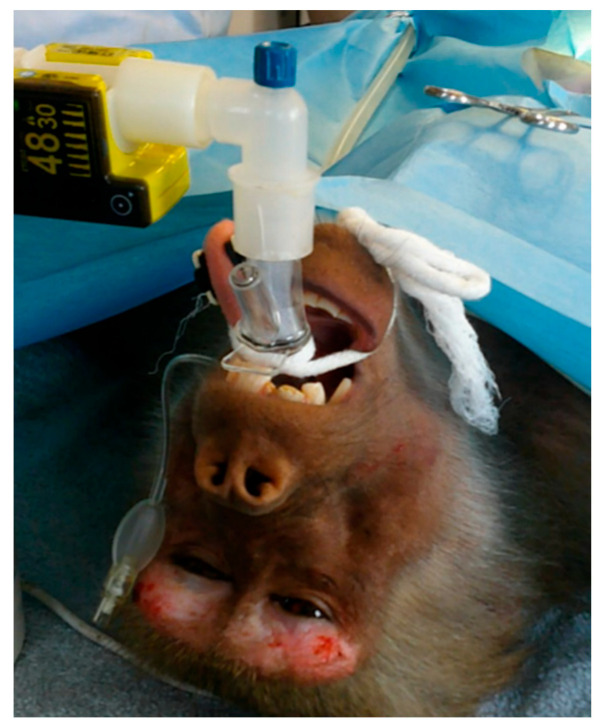
One of the baboons involved in the study after the insertion of the LMA device. The mainstream capnograph is connected to the LMA using an L connector.

**Figure 3 vetsci-10-00158-f003:**
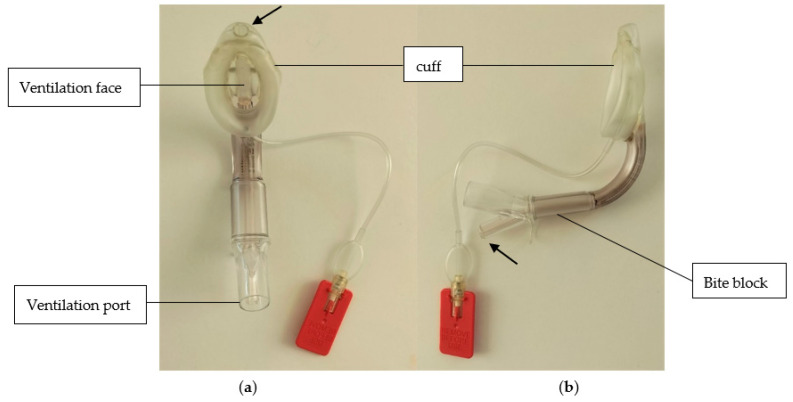
(**a**) Frontal and (**b**) lateral views of the laryngeal mask, LMA Supreme™, used in this study. The narrows indicate the tube for the drainage of the stomach (**a**) and the gastric access port (**b**).

**Table 1 vetsci-10-00158-t001:** Complete blood count (CBC) (mean and 95% confidence interval, CI) in 11 baboons (*Papio hamadryas*) 15 min after intramuscular injection with tiletamine-zolazepam and medetomidine.

Variable	Mean (95% CI)	Reference Range ^a^
RBC (M/µL)	4.7 (4.3–5.1)	3.9–6.9
Hb (g/dL)	11.7 (10.7–12.7)	9.1–17.5
Hct (%)	36.3 (33.9–38.8)	30.3–61.1
MCV (fL)	77.2 (75.8–78.6)	64.3–97.4
MCHC (g/dL)	32.4 (31.1–33.8)	24.6–40.2
WBC (K/µL)	9.2 (6.6–11.7)	3.2–28.6
NEU (/µL)	7200 (4700–9800)	1090–24,300
LYM (/µL)	1300 (900–1800)	1290–8760
MONO (/µL)	500 (300–600)	0–2106
EOS (/µL)	20 (7–40)	0–923
BASO (/µL)	30 (20–40)	0–172
Plt (K/µL)	448.9 (401.8–495.9)	157–875

RBC, red blood cells; Hb, hemoglobin; Hct, hematocrit; MCV, mean cell volume; MCHC, mean corpuscular hemoglobin concentration; WBC, white blood cells; NEU, neutrophils; LYM, lymphocytes; MONO, monocytes; EOS, eosinophils; BASO, basophils; Plt, platelets. ^a^ International Species Information System (ISIS) [23].

**Table 2 vetsci-10-00158-t002:** Biochemical profile (mean and 95% CI) in 11 baboons (*Papio hamadryas*) 15 min after intramuscular injection with tiletamine-zolazepam and medetomidine.

Variable	Mean (95% CI)	Reference Range ^a^
CPK (IU/L)	626.7 (38.6–1214.7)	38–2262
AST (IU/L)	45.2 (34.4–56.1)	11–141
ALT (IU/L)	29.5 (16.8–42.2)	11–107
ALP (IU/L)	437.7 (204.4–670.9)	55–1213
GGT (IU/L)	35.3 (26.9–43.7)	13–123
TB (mg/dL)	0.17 (0.14–0.2)	0–1.0
TP (g/dL)	6.3 (5.8–6.8)	5–8.9
Alb (g/dL)	3.4 (2.9–3.9)	2.8–5.2
Glob (g/dL)	2.9 (2.6–3.2)	1.6–5.1
Chol (mg/dL)	97.5 (80.1–115.1)	55–189
Trig (mg/dL)	49.9 (37.6–62.1)	14–149
BUN (mg/dL)	31.1 (21.6–40.5)	6–29
Crea (mg/dL)	1.1 (0.8–1.2)	0.6–1.9
Gluc (mg/dL)	134.1 (104.5–163.6)	37–391
Ca (mg/dL)	8.9 (8.3–9.4)	7.8–11.4
P (mg/dL)	3.6 (2.4–4.7)	0.8–9.5
Mg (mg/dL)	1.7 (1.5–1.9)	1.1–2.2
Na (mEq/L)	145.3 (142.6–147.9)	141–162
K (mEq/L)	4.1 (3.5–4.5)	2.8–5.1
Cl (mEq/L)	106.1 (103.1–109.2)	96–126

CPK, creatine phosphokinase; AST, aspartate aminotransferase; ALT, alanine aminotransferase; ALP, alkaline phosphatase; GGT, gamma glutamyl transferase; TB, total bilirubin; TP, total protein; Alb, albumin; Glob, globulin; Chol, cholesterol; Trig, triglyceride; BUN, blood urea nitrogen; Crea, creatinine; Gluc, glucose; Ca, calcium; P, phosphorus; Mg, magnesium; Na, sodium; K, potassium; Cl, chloride. ^a^ International Species Information System (ISIS) [23].

**Table 3 vetsci-10-00158-t003:** Mean values and 95% CI of the values of heart rate (HR), mean arterial pressure (MAP), respiratory rate (RR), end tidal carbon dioxide concentration (EtCO_2_), tidal volume (TV), minute volume (MV), and peripheral hemoglobin oxygen saturation (SpO_2_) of the animals included in the study.

Parameters	PREPP	PP1	PP2	PP3	PP4	POSTPP
HR (beats/min)	64 (57–71)	58 (50–66) *	60 (55–66)	61 (56–67)	61 (55–68)	61 (55–68)
MAP (mmHg)	82 (72–93)	73(59–88)	85(69–101) °	84 (65–103)	84 (64–103)	82 (65–99)
RR (breaths/min)	17 (13–21) #	18 (11–25) *#	20 (14–26) *#	21 (16–27) *#	23 (18–28) *	21 (16–27) *
EtCO_2_ (mmHg)	44 (41–48)	43 (39–48)	55 (51–60) *°	58 (53–62) *°	58 (52–64) *°	53 (50–57) *°
TV (mL/kg)	11.7 (7.2–16.1)	10.4 (6.8–13.9)	11.3 (7.8–14.7)	11.1 (8.1–14.0)	11.4 (8.3–14.4)	12.6 (10.2–14.9)
MV (mL/min/kg)	213.5 (158.7–268.3)	184.4 (146.1–221.9)	198.3 (161.9–234.8)	207.3 (168.1–246.7)	249.4 (190.7–308.1) §	269.1 (206.1–331.8) §
SpO_2_ (%)	97 (96- 98)	97 (96–99)	98 (97–99)	98 (96–99)	98 (98–99)	98 (98–99)

* = *p* < 0.05 compared to PREPP. # = *p* < 0.05 compared to PP4. ° = *p* < 0.05 compared to PP1. § = *p* < 0.05 compared to PREPP, PP1, PP2, and PP3.

## Data Availability

Not applicable.

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
