# Peer review of "Use of Laryngeal Mask and Anesthetic Management in Hamadryas Baboons (Papio hamadryas) Undergoing Laparoscopic Salpingectomy—A Case Series"

_vetsci, 2023, doi:10.3390/vetsci10020158_

Round 1

Reviewer 1 Report

Dear Authors, 

I reviewed your manuscript. In my opinion the manuscript lacks clarity regarding its objectives and its approach. There are also some inconsistencies that would need to be addressed. My comments can be found below, and I hope you will find them useful for improving your manuscript. 

L65-72: In this section, you are stating that most of the data regarding anaesthesia in Baboon is from experimental purpose. However, for the cited reference, the results can be used for clinical purpose. Regarding the laparoscopy procedure, you are stating that cardiovascular and respiratory effects, in a clinical scenario, are missing; however, the surgery in this study has been done in an elective context very similar to surgical procedure for experimental purpose. Overall, this section seems to underestimate the quality of anaesthesia data obtained from laboratory animal setting. 

L84: The objectives of the study are not clear. 

L85: This information was already mentioned. The choice of the procedure could be more justified

L107: The doses must be in mg/kg. 

L127: How did you perform the leak test?

L135: RR and bpm; check this is the right unit 

L138: where was the anatomical location of the BP cuff

L145: space

L154: has fentanyl been administered on any occasion?

L173: the use of the t-test in this section is incorrect as the value are not independent but all linked over the time. A repeated ANOVA will be the most appropriate statistical test in this occasion. This must be reviewed.

Also, if this is an observational study it is not clear what the statistical analysis is bringing. Please state the statistical hypothesis you are conducting.

L216: The results section must be reviewed considering the correct statistical test

L225: This statement would be better placed at the beginning of the results section

L274: Were the animals anaesthetised deeply enough for the procedure? What was the amplitude of changes (HR and MAP). Was fentanyl administered at this point?

L279: typo “pneumoperitoneum”

L291: About this statement; you are again making a clear point to differentiate data from research setting and clinical practice. But data from laboratory animal studies are often cleaner than clinical work and as a result bring a wealth of information? In which capacity those are different?

L300: When did the administration of oxygen start? Please clarify 

L318: This statement is incorrect. It is not the placement of the LMA that improved the SpO2 but the provision of a FiO2 above 0.8

L320: Again, this is incorrect you cannot interpret the EtCO2 results isolated from the other respiratory parameters. The animals were spontaneously breathing and sometime were manually ventilated.

L333: Is this really a frequent issue? This statement must be backed up with a reference. More importantly this could be a technic used by person less experienced in intubation like in case of emergency

L358: Confusing why are you looking into the anaesthesia protocol? This was not clearly stated as an objective

L361: you cannot conclude on the ventilation without blood gas data. Again, animals are in spontaneous breathing

Author Response

Dear Authors, 

I reviewed your manuscript. In my opinion the manuscript lacks clarity regarding its objectives and its approach. There are also some inconsistencies that would need to be addressed. My comments can be found below, and I hope you will find them useful for improving your manuscript. 

Dear Reviewer, we would like to thank you for the time dedicated to the revision of our manuscript. We appreciated your comments and we have tried to amend the article based on your suggestions. 

L65-72: In this section, you are stating that most of the data regarding anaesthesia in Baboon is from experimental purpose. However, for the cited reference, the results can be used for clinical purpose. Regarding the laparoscopy procedure, you are stating that cardiovascular and respiratory effects, in a clinical scenario, are missing; however, the surgery in this study has been done in an elective context very similar to surgical procedure for experimental purpose. Overall, this section seems to underestimate the quality of anaesthesia data obtained from laboratory animal setting. 

Dear reviewer we would like you to better consider the papers we are referencing. They are pure experimental studies with specific, very invasive procedure which do not provide any type of clinical information. We have improved the test of the manuscript to make it more clear.

L84: The objectives of the study are not clear. 

The text has been modified to make it more clear

L85: This information was already mentioned. The choice of the procedure could be more justified

We agree and we have removed the sentence. In the aim of the study we should not justify the surgical procedure

L107: The doses must be in mg/kg. 

Done

L127: How did you perform the leak test?

The leak test was performed closing the apl valve of the breathing circuit, turning off the oxygen flow and squeezing the reservoir bag until reaching 20 cmH2O of airway pressure and checking for the absence of gross air leak (hearing gas flow and/or smelling isoflurane).

 We have added this info in the manuscript

L135: RR and bpm; check this is the right unit 

Done

L138: where was the anatomical location of the BP cuff

The cuff was positioned on the antebrachium, at the same level of the heart, and the size corresponded to the 40-50% of the circumference of the arm.

L145: space

Done 

L154: has fentanyl been administered on any occasion?

None of the animals received rescue doses of fentanyl. This information was already reported in the first version of the manuscript: None of the animals required the rescue dose of fentanyl

L173: the use of the t-test in this section is incorrect as the value are not independent but all linked over the time. A repeated ANOVA will be the most appropriate statistical test in this occasion. This must be reviewed.

Also, if this is an observational study it is not clear what the statistical analysis is bringing. Please state the statistical hypothesis you are conducting.

Dear reviewer the one-way anova test has been applied and the results didn’t change in terms of statistical difference. The reason to run an anova test was to see if there was any variation in relation to the pneumoperitoneum. This has been now specified in the text

L216: The results section must be reviewed considering the correct statistical test

As we mentioned before the results didn’t change

L225: This statement would be better placed at the beginning of the results section

Done

L274: Were the animals anaesthetised deeply enough for the procedure? What was the amplitude of changes (HR and MAP). Was fentanyl administered at this point?

We have reported the criteria for rescue analgesia and as we reported none of the cases required it. For Surgical stimulation we mean the basic sympathetic activation that in same case could be related to the surgical manipulation. None of the cases required rescue analgesia and the level of anesthesia was adequate in all animals. We have reported this information in the manuscript

L279: typo “pneumoperitoneum”

Done

L291: About this statement; you are again making a clear point to differentiate data from research setting and clinical practice. But data from laboratory animal studies are often cleaner than clinical work and as a result bring a wealth of information? In which capacity those are different?

We have replied to this question previously

L300: When did the administration of oxygen start? Please clarify 

We have modified the text as follow:

However, low SpO2 at room air was a common finding (88-92%), that in our study was solved by O2 supplementation (FiO2>0.8) that started after the LMA was correctly inserted. T

L318: This statement is incorrect. It is not the placement of the LMA that improved the SpO2 but the provision of a FiO2 above 0.8

Modified

L320: Again, this is incorrect you cannot interpret the EtCO2 results isolated from the other respiratory parameters. The animals were spontaneously breathing and sometime were manually ventilated.

We have modified the text as follow: Even the EtCO2 increased significantly during pneumoperitoneum and remained high even after. Anyway, the rise of EtCO2 was not clinically severe and laid below the threshold value for starting manually assisted ventilation.

L333: Is this really a frequent issue? This statement must be backed up with a reference. More importantly this could be a technic used by person less experienced in intubation like in case of emergency

We have modified the text as follow: The easy of insertion of LMA could play an important role in the spreading of this kind of device in emergencies settings and among less experienced professionals. Moreover, in the anesthesia of NHPs the spread of use of the supraglottic devices can decrease the risk for monobronchial intubation due to the shorter trachea compared to humans.

L358: Confusing why are you looking into the anaesthesia protocol? This was not clearly stated as an objective

We modified the word protocol in management 

L361: you cannot conclude on the ventilation without blood gas data. Again, animals are in spontaneous breathing

We have removed the words gas exchange but we left ventilation because this function can be adequately estimate by capnography

Reviewer 2 Report

The article is well structured, its reading is pleasant and it provides substantial information for the management of the airway in the species studied.

The present study aims to describe the anesthesiological management of eleven baboons in which an LMA has been used for airway management when they underwent laparoscopic salpingectomy for birth control in a zoo.

The topic is original and provides a clinical approach to airway management during laparoscopic surgery.

It provides new information not yet reported in the literature on the animal species studied.

The methodology is adequate and the limitations of the study have been well identified and justified. 

The conclusions are consistent with the results and the authors provide additional information on advanced devices to reduce the incidence of regurgitation. The main question posed is adequately answered.

The references are adequate in quality, relevance and quantity.

In the line 145 where it says “mg/kg IV and cefazoline” it may say “mg/kg IV and cefazoline”.

Author Response

Dear reviewer,

thank you for your comments. We appreciate that you enjoyed the article.

The grammatical error you noticed has been corrected (line 145: All animals received as preoperative standard treatment meloxicam (Metacam, Boehringer Ingelheim,Milan, Italy) 0.2 mg/kg IV and cefazoline (Cefazolina Teva, Teva Italia S.r.l., Milan, Italy) 20 mg/kg  IV.).

Reviewer 3 Report

Thank-you for this interesting case series. 

I have few comments: 

- in presenting results, please could you report to the accuracy of measurement (e.g. respiratory rate, should be a whole number, blood pressure to the nearest whole mmHg...)

- English is good in introduction and methods but could be improved particularly in the discussion. 

- This is a case series of a specific anaesthetic protocol without a comparator - It is original paper - Use of laryngeal masks have not been reported in this species - Ideally a control or comparator could have been included, but this is not appropriate in this clinical situation- but perhaps could be clearer by indicating 'a case series' in the title. - conclusions and arguments are appropriately made - references appear appropriate - I am not sure how necessary the photograph is, but the table is clear and easy to read.

Author Response

Dear reviewer, thank you for your interesting and useful comments. We started from them to try to improve the quality and the clarity of the manuscript, as you can read below.

- in presenting results, please could you report to the accuracy of measurement (e.g. respiratory rate, should be a whole number, blood pressure to the nearest whole mmHg...)

 data of MAP and RR in the results have been replaced with whole numbers, as well as those of HR, ETCO2 and SpO2;

- English is good in introduction and methods but could be improved particularly in the discussion.         

 We have worked to improve the quality of English in the discussion;

- This is a case series of a specific anaesthetic protocol without a comparator - It is original paper - Use of laryngeal masks have not been reported in this species - Ideally a control or comparator could have been included, but this is not appropriate in this clinical situation- but perhaps could be clearer by indicating 'a case series' in the title. – conclusions and arguments are appropriately made - references appear appropriate - I am not sure how necessary the photograph is, but the table is clear and easy to read.

 We added “- a case series” in the title as you suggested.

Furthermore, we have written a paragraph with limitations in discussion section where it is discussed the lack of a group with endotracheal tube for comparison: “Among the limitations of the study, there are the lack of comparison with endotracheal intubation and the wide range of dose of the sedative drugs, that could have influenced physiological parameters throughout the procedure.” (L372-374).

Reviewer 4 Report

General comments- Thank you for sharing this work. As the authors mention, publications on anesthetic techniques in baboons and other NHPs are limited, making the information presented here of great interest to individuals that care for NHP. A comparative study of endotracheal tube vs LMA in baboons may have been stronger scientifically, but shouldn't necessarily exclude this from consideration for publication. A more detailed discussion of potential study weaknesses would be helpful. One example includes the wide range of initial injectable TZM and how this could have potentially influenced values throughout the procedure. Please see additional comments related to specific sections below:

Line 50-51: Are there more references to support this claim? This is not convincing with a mention of a single species with stress evaluated based on blood values.

Line 59: remove word "because" 

Line 94: Is age known for any/all animals?

Line 103-108: Please include concentration (mg/ml) of the drugs used so the reader can extrapolate total mg used.

Line 118: How much propofol was given? Was the volume needed recorded?

Line 131-132: What range of isoflurane % was used/needed?

Line 145: "andcefazoline" should this be changed to "and cefazolin"?

Results: General comment- showing the LMA size used and weight for each animal would be helpful to the reader. Did it match the human weight/size guidelines presented in the methods?

Line 225: "any case" should probably be changed to "all cases"

Discussion: General comment- There is no discussion of the CBC/Chem results or what significance this data contributes to the manuscript. A single time point is represented. Is it presented to show health status of the animals? 

Line 291-292: "baboons are referred to" may read better as "baboons refer to"

Line 308-310: Use also reported in humans right? 

Line 312-315: This should probably be made into two sentences and reworded to improve clarity.

Line 319: The change in SpO2 was not statistically significant correct? This should be mentioned with any claim of "improving oxygenation". The 93.1% pre SpO2 value does not match the value presented in the results table (97.3)

Line 320-321: The statistical analysis in the results show that EtCO2 did increase significantly, correct? Perhaps you mean to say that the increase is not clinically severe/signficant/concerning?

Line 323-326: Do these references claim and show data to support that all of these parameters are improved in LMA as compared to ET tube? LMA allows better maintenance of oxygenation and ventilation? and reduces nausea? Was this in a specific setting or procedure? These sound like strong claims that could lead the reader into thinking an LMA is always better than ETT.

Author Response

Dear reviewer, thank you for your interesting and useful comments. We started from them to try to improve the quality and the clarity of the manuscript, as you can read below.

General comments- Thank you for sharing this work. As the authors mention, publications on anesthetic techniques in baboons and other nhps are limited, making the information presented here of great interest to individuals that care for NHP. A comparative study of endotracheal tube vs LMA in baboons may have been stronger scientifically, but shouldn't necessarily exclude this from consideration for publication. A more detailed discussion of potential study weaknesses would be helpful. One example includes the wide range of initial injectable TZM and how this could have potentially influenced values throughout the procedure.

  • We have added a paragraph in the discussion section with a list of limitations/weaknesses: “Among the limitations of the study, there are the lack of comparison with endotracheal intubation and the wide range of dose of the sedative drugs, that could have influenced physiological parameters throughout the procedure” (lines 372-374)

Please see additional comments related to specific sections below:

Line 50-51: Are there more references to support this claim? This is not convincing with a mention of a single species with stress evaluated based on blood values.

  • More references have been added to support this claim, so the range of species has been expanded too: “Perioperative systemic stress response, as assessed by the determination of metabolic and endocrine indices, is significantly reduced in the laparoscopic surgery compared to open surgery [4-6]”

References:

[4]       A. Krikri; V. Alexopoulos; E. Zoumakis; P. Katsaronis, E. Balafas; G. Kouraklis; P. Karayannakos; G. Chrousos; G. Skalkeas. “Laparoscopic vs. Open abdominal surgery in male pigs: Marked differences in cortisol and catecholamine response depending on the size of surgical incision.”, Hormones, 2013, vol. 12(2), 283-291.

[5]       Devitt CM, Cox RE, Hailey JJ. Duration, complications, stress, and pain of open ovariohysterectomy versus a simple method of laparoscopic-assisted ovariohysterectomy in dogs. J Am Vet Med Assoc. 2005 Sep 15;227(6):921-7. Doi: 10.2460/javma.2005.227.921. PMID: 16190590.

[6]       Haleem, Shahla; Mohd. Ansari, Maulana; Mussarat, Jawed; Ahmed, Aftab; Islam, Najmul; Bano, S.; Singh, Braj Raj. Cortisol and Glycemic Response to Open and Laparoscopic Cholecystectomy - A Comparative Evaluation. Journal of Anaesthesiology Clinical Pharmacology 24(4):p 437-440, Oct–Dec 2008.

Anyway, the stress response is evaluated on metabolic and endocrine indices since they are the most used indicators of perioperative stress response;

Line 59: remove word "because" 

  • Done: “These are the main reasons for which adequate airway management and a close monitoring are of great importance during laparoscopy” (L58-60);

Line 94: Is age known for any/all animals?

  • Unfortunately, we don't know the exact age of each animal;

Line 103-108: Please include concentration (mg/ml) of the drugs used so the reader can extrapolate total mg used.

  • Done: “The animals were restrained in a squeeze cage and hand injected with tiletamine-zolazepam (Zoletil 100 mg/ml, Virbac S.r.l., Milan, Italy) and medetomidine (Domitor 1 mg/ml, Zoetis s.r.l., Rome, Italy) IM combined in the same syringe” (L103-105);

Line 118: How much propofol was given? Was the volume needed recorded?

  • The volume of propofol administered for induction was not recorded in all cases;

Line 131-132: What range of isoflurane % was used/needed?

  • Isoflurane vaporizer was set to 1,0-1,5%. This information has been added in the text: “Animals were connected to a rebreathing circuit in spontaneous ventilation and isoflurane (isoflo®, Zoetis Italia S.r.l., Rome, Italy) in oxygen was administered for maintaining anesthesia. Vaporizer was set to 1.0-1.5%.” (L131-133);

Line 145: "andcefazoline" should this be changed to "and cefazolin"?

  • Done;

Results: General comment- showing the LMA size used and weight for each animal would be helpful to the reader. Did it match the human weight/size guidelines presented in the methods?

  • Accordance between LMA size-weight of the animal has been successfully get using manufacturer’s recommendation. This is now specified: “The adequacy of the size of LMA to the laryngeal opening of the baboons suggests that manufacturer’s recommendations for human patients are valid also for this species” (LL 257-259);

Line 225: "any case" should probably be changed to "all cases"

  • Done: “The LMA insertion was successful in all cases and no animal required intubation” (L 226):

Discussion: General comment- There is no discussion of the CBC/Chem results or what significance this data contributes to the manuscript. A single time point is represented. Is it presented to show health status of the animals? 

  • CBC and biochemical results have been briefly commented: “Hematological values are within the reference range reported for the species, unless the BUN, that is slightly higher. Although the data show the situation at a single time point, they are useful to verify the health status of the animals, as well as hepatic and renal functions and the CBC” (lines 265-268);

Line 291-292: "baboons are referred to" may read better as "baboons refer to"

  • Done: “In literature, all protocols reported for anesthetic management of baboons refer to experimental studies [10-12].” (L 299-300);

Line 308-310: Use also reported in humans right? 

  • We have now specified that references are for veterinary medicine: “Laryngeal mask was introduced in human medicine in 1980s from an idea of Dr. Brain [26]. In veterinary medicine its use has been reported in dogs [27], cats [28], rabbits [29], pigs [30], capibaras [31], and in nhps (chimpanzees, gibbons, and gorillas) [20-21]”(L317-319). Articles with LMA in human medicine are reported in the paragraph below;

Line 312-315: This should probably be made into two sentences and reworded to improve clarity.

  • Done: “Cerveny compared endotracheal tube (ETT) with LMA in gorillas and found higher spo2 and partial pressure of oxygen (pao2) and lower RR in LMA group. However, the treatment group was not randomized and animals with greater risk of respiratory complications were included in group ETT, so this may have accounted for the observed differences.” (321-325);

Line 319: The change in spo2 was not statistically significant correct? This should be mentioned with any claim of "improving oxygenation". The 93.1% pre spo2 value does not match the value presented in the results table (97.3)

  • The statement has been modified in order to improve clarity: “spo2 increased from pre-anesthetic spat (93%, 95% CI 88 - 98) when oxygen started to be administered through LMA (97-98% during the times of the study)” (lines 329-331).

 In fact spo2 93% refers to the peripheral oxygen saturation measured prior to the LMA placement and the administration of oxygen. This value is reported in results but not in the table. The spo2 reported in the table refers to the saturation during surgery, that started after the LMA placement;

Line 320-321: The statistical analysis in the results show that etco2 did increase significantly, correct? Perhaps you mean to say that the increase is not clinically severe/signficant/concerning?

  • Yes, we do. In fact, the sentence has been modified as follows:” Even the etco2 increased significantly during pneumoperitoneum and remained high even after. Anyway, the rise of etco2 was not clinically severe and laid below the threshold value for starting assisted ventilation.” (L 332-334);

Line 323-326: Do these references claim and show data to support that all of these parameters are improved in LMA as compared to ET tube? LMA allows better maintenance of oxygenation and ventilation? And reduces nausea? Was this in a specific setting or procedure? These sound like strong claims that could lead the reader into thinking an LMA is always better than ETT.

  • The statement has been modified based on your observation and refers to a more restricted selection of patients: “In humans, LMA apported some advantages over the endotracheal tube in elective laparoscopic procedures and in patients without specific requirements or diseases: maintenance of adequate oxygenation and ventilation perioperatively, minor incidence of postoperative pharyngolaryngeal morbidity, from minimum to zero incidence of hemodynamic stress response and minor rates of postoperative nausea and vomiting [32-33].” (L336-340).